# Dysfunctional Chondroitin 4-*O*-Sulfotransferase-1 Impairs Cellular Redox State and Promotes Tau Aggregation

**DOI:** 10.3390/cells14211686

**Published:** 2025-10-28

**Authors:** Satomi Nadanaka, Yuto Imamoto, Toru Takarada, Masafumi Tanaka, Hiroshi Kitagawa

**Affiliations:** 1Laboratory of Biochemistry, Kobe Pharmaceutical University, Higashinada-ku, Kobe 658-8558, Japan; snadanak@kobepharma-u.ac.jp (S.N.); yt.imamoto1939@outlook.jp (Y.I.); 2Laboratory of Functional Molecular Chemistry, Kobe Pharmaceutical University, Higashinada-ku, Kobe 658-8558, Japan; takarada@kobepharma-u.ac.jp (T.T.); masatnk@kobepharma-u.ac.jp (M.T.)

**Keywords:** proteoglycan, chondroitin sulfate, oxidative stress, glutathione, tau aggregation

## Abstract

**Highlights:**

**What are the main findings?**

**What is the implication of the main finding?**

**Abstract:**

Chondroitin sulfate (CS) chains on the cell surface are sulfated in various patterns, and this structure is the basis of CS function. We aimed to investigate the role of chondroitin 4-*O*-sulfotransferase-1 (C4ST-1), the enzyme responsible for the 4-sulfation of CS, in redox homeostasis and protein aggregation in mouse neuroblastoma Neuro2a and neural progenitor C17.2 cells. Results showed that C4ST-1 deficiency significantly reduced 4-sulfated CS, which led to markedly decreased intracellular glutathione levels and increased reactive oxygen species production. Mechanistically, C4ST-1 loss reduced the CS modification of neurocan, decreased the stability of the cystine transporter xCT, and decreased intracellular glutathione levels. This redox imbalance promoted protein aggregation and caused lysosomal membrane damage, indicating a failure of protein quality control. Although C4ST-1 deficiency alone did not cause tau protein aggregation, it significantly accelerated the aggregation of a familial tauopathy mutant following the introduction of seeds. These findings suggest that C4ST-1-mediated CS sulfation regulates the intracellular redox state and tau pathology. Thus, C4ST-1 may serve as a therapeutic target for neurodegenerative diseases.

## 1. Introduction

Oxidative stress is crucial in the onset and progression of neurodegenerative diseases, such as Alzheimer’s disease (AD) [1]. Evidence shows the accumulation of oxidative stress in the brains of patients with AD and model mice. Mild cognitive impairment is also marked by increased levels of oxidized proteins and lipids [2].

Glutathione (GSH), a tripeptide composed of glutamate, cysteine, and glycine, is an important neuroprotective substance in the central nervous system. Its dysregulation has been associated with the development of neurodegenerative diseases [3]. Cellular cysteine uptake, which is mainly conducted by excitatory amino acid carrier 1 (EAAC1), is the rate-limiting step for the production of GSH in neurons; EAAC1 dysfunction decreases GSH levels in the brain and affects neurodegeneration [4]. The cystine-glutamate reverse transporter xCT (SLC7A11) is expressed and functions in certain neuronal cell types under pathological conditions, such as neuroinflammation and oxidative stress [5]. These findings suggest that neuronal GSH homeostasis is dynamically regulated by multiple transporters and that the dysfunction of these transporters is involved in oxidative stress-mediated pathologies.

Chondroitin sulfate (CS) is another substance gaining attention for its antioxidant properties. It can directly neutralize reactive oxygen species (ROS), chelate metal ions [6], and enhance the endogenous antioxidant system by upregulating Nrf2, a master regulator of antioxidant responses [7]. These protective functions make CS a promising drug target for neurodegenerative diseases.

The function of CS is highly dependent on its sulfation pattern. Reduced levels of 4-*O*-sulfated CS due to defective C4ST-1 expression increase susceptibility to oxidative stress [8]. This finding suggests that C4ST-1-synthesized CS plays a conserved role in oxidative stress protection.

Considering these previous studies, we hypothesized that CSs synthesized by C4ST-1 play an important role in redox state regulation in neurons. We further hypothesized that C4ST-1 deficiency leads to an oxidative state, which promotes pathological protein aggregation such as that of tau. This study aims to use a neuronal cell model to elucidate a new pathological mechanism linking altered redox state and tau aggregation.

## 2. Materials and Methods

### 2.1. Cell Culture and C4ST-1 Knockout Cell Generation

The mouse neuroblastoma cell line Neuro-2a (IFO50081) and C17.2 (Cat. No. 07062902) was obtained from the Japanese Collection of Research Bioresources and the European Collection of Cell Cultures, respectively. Neuro-2a cells were grown in Eagle’s minimal essential medium supplemented with non-essential amino acids, 10% heat-inactivated fetal bovine serum, 100 units/mL penicillin, 100 mg/mL streptomycin, and 1% GlutaMAX^TM^ supplement. C17.2 cells were grown in Dulbecco’s modified Eagle medium supplemented with 5% heat-inactivated horse serum, 10% heat-inactivated fetal bovine serum, 100 units/mL penicillin, and 100 mg/mL streptomycin. All cell lines were maintained at 37 °C and 5% CO_2_. Information regarding the reagents used for cultivation is shown in Appendix A.

CRISPR/Cas9 plasmids targeting the mouse *C4st-1* gene using a GeneArt^®^ CRISPR Nuclease Vector with orange fluorescent protein reporter kit (#A21174) (Thermo Fisher Scientific, Waltham, MA, USA) were constructed as described previously (Appendix A) [9].

### 2.2. Disaccharide Analysis of GAGs from N2A Cells

CSs isolated and purified from N2A cells and their clones were analyzed as described previously [10].

### 2.3. Real-Time PCR

Total RNA was isolated from cells and cDNA was synthesized as described previously [10]. Quantitative real-time PCR was conducted using THUNDERBIRD^®^ Probe qPCR Mix in a LightCycler 96^®^ (Roche Diagnostics GmbH, Vienna, Austria) in accordance with the manufacturer’s protocols. The primers used for real-time PCR are shown in Appendix A.

### 2.4. Knockdowns

The siRNA-mediated gene silencing of *Ncan* (mm.Ri.Ncan.13.1 and mm.Ri.Ncan.13.2 Integrated DNA Technologies, Inc., Coralville, IA, USA) and *Sdc3* (mm.Ri.Sdc3.13.1 and mm.Ri.Sdc3.13.2 Integrated DNA Technologies, Inc.) (Appendix A) was performed as described previously [11].

### 2.5. ROS and Protein Aggregate Detection by FACS

ROS generated in each cell were labeled with the ROS assay kit—Highly Sensitive DCFH-DA- (#R252, Dojindo Laboratories, Kumamoto, Japan) in accordance with the manufacturer’s instructions. Protein aggregates within the cells were labeled with the ProteoStat^®^ Aggresome detection kit (#ENZ-51035, ENZO Life Sciences, Long Island, NY, USA) in accordance with the manufacturer’s instructions. The labeled cells were analyzed using the BD Accuri^TM^ C6 flow cytometer (BD, Franklin Lakes, NJ, USA).

### 2.6. Immunoblotting and Immunoprecipitation

Immunoprecipitation was performed using the antibodies listed in Appendix A as described previously [9]. Immunoblotting was carried out using the primary antibodies listed in Appendix A as described previously [10]. Whole Western blots that show all bands with all molecular weight markers are shown in Appendix A.

### 2.7. Immunofluorescence

Cells were grown until 80%–90% confluent on a CELLview four-chamber glass-bottomed dish (Cat. No. 627870, Greiner Bio-One, Kremsmunster, Austria) and then stained with the following antibodies as described previously [9]: anti-galectin 3 antibody (dilution ratio 1:100; Cat. No. 14979-1-AP, Proteintech, Rosemont, IL, USA) and anti-LAMP-1 antibody (dilution ratio 1:100; Clone H4A3, Material Number 555798, BD Biosciences, Franklin Lakes, NJ, USA). Hoechst33342 was used as a nuclear counter stain. Images were acquired with a Zeiss LSM700 confocal laser scanning system (Carl Zeiss Inc., Oberkochen, Germany) equipped with an inverted Axio observer Z1 microscope.

### 2.8. MTT Assay

Cells were treated with 10 nM bafilomycin A1 (#54645, Cell Signaling Technology, Danvers, MA, USA) for 24 h. MTT assay was performed using the MTT cell count kit (#23506-80, Nacalai tesque, Kyoto, Japan).

### 2.9. Modification of Free Thiol Groups on a Tau Protein

The free -SH groups were labeled with Protein-Shifter Plus (#SB12, Dojindo Laboratories) in accordance with the manufacturer’s protocol.

### 2.10. Expression of Mutant Tau Protein and Detection of Tau Aggregation

Mutant tau protein aggregation in N2A and C4ST-1KO cells was compared using the Tau aggregation assay kit (#TAU01, Cosmo Bio Co., Ltd., Tokyo, Japan) in accordance with the manufacturer’s instructions. Cells expressing Tau(2N4R)-P301L with or without 4R-Tau (P301L) fibrils (tau seeds) were solubilized and fractionated.

### 2.11. Statistical Analysis

Data are expressed as the mean ± standard deviation of the mean. Statistical significance was determined using Student’s *t*-test and one-way ANOVA followed by Tukey’s or Dunnett’s multiple comparison test.

## 3. Results and Discussion

### 3.1. Regulation of Reduced Glutathione Levels in Neuro2a and C17.2 Cells by C4ST-1 Expression

The CS chain consists of a repeating disaccharide unit composed of a glucuronic acid residue and an *N*-acetylgalactosamine (GalNAc) residue (Figure 1a). The unsulfated disaccharide is called an O unit, which is converted to an A unit by sulfation of the GalNAc residue at position 4 by C4ST-1. To evaluate the effect of C4ST-1 function on the sulfation pattern of CS, we generated Neuro2a (N2A) cells lacking C4ST-1 by genome editing (C4ST-1KO cells) and cells in which C4ST-1 was re-expressed (C4ST-1KO_C4ST-1 cells). Analysis of disaccharide composition showed a significant decrease in 4-sulfated A units and a significant increase in unsulfated O units in C4ST-1KO cells compared with N2A cells (Figure 1b). In addition, C4ST-1KO_C4ST-1 cells showed an increase in A units and a significant decrease in O units compared with C4ST-1KO_empty cells, confirming that C4ST-1 is the major enzyme responsible for 4-sulfation in N2A cells. Subsequently, the effect of CS 4-sulfation by C4ST-1 on the intracellular redox state of N2A, C4ST-1KO, and C4ST-1KO_C4ST-1 cells was examined by measuring the intracellular GSH concentration; compared with N2A cells, C4ST-1KO cells had significantly lower GSH concentration, and a trend toward the recovery of GSH concentration was observed in C4ST-1KO_C4ST-1 cells compared with C4ST-1KO_empty cells (Figure 1c). In C17.2 cells, *C4st-1* knockout resulted in a significant decrease in A units and a significant increase in O units (Figure 1d). Furthermore, the intracellular GSH concentration was significantly reduced in C4ST-1 KO cells (Figure 1e). Whether this decrease in GSH concentration leads to increased oxidative stress was evaluated. Results showed that ROS levels were significantly elevated in C4ST-1KO cells compared with N2A cells (Figure 1f). As observed in N2A cells, C4ST-1 deficiency also led to increased ROS in C17.2 cells (Figure 1f). These results clearly indicate that reduced 4-sulfation of CS due to C4ST-1 dysfunction leads to a decrease in intracellular GSH concentration and a concomitant increase in oxidative stress. This phenomenon suggests that the specific sulfation pattern of CS helps regulate the cellular antioxidant defense system.

### 3.2. Regulation of GSH Concentration in N2A Cells by Proteoglycans

CS is present on the cell surface and in the extracellular matrix as proteoglycans (PGs) bound to core proteins; PGs expressed at relatively high levels in N2A cells and modified with CS are neurocan (NCAN) and syndecan-3 (SDC3). We examined whether these PGs are involved in the regulation of intracellular GSH levels. We suppressed the expression of these PGs using siRNAs. Two siRNAs against *Ncan* (siNcan#1, siNcan#2) significantly reduced *Ncan* expression to approximately 20% (Figure 1g) and intracellular GSH levels (Figure 1h). We then performed the same experiment using two siRNAs against *Sdc3* (siSdc3#1, siSdc3#2). These siRNAs reduced the expression of *Sdc3* to approximately 20%, but the effect on the intracellular GSH concentration was significant only with siSdc3#2 and not with siSdc3#1 (Figure 1g,h). These results strongly suggest that NCAN helps regulate intracellular GSH concentration in N2A cells.

### 3.3. Regulation of Intracellular GSH Concentration via PGs and xCT

xCT is primarily expressed in astrocytes, serving as a major source of brain GSH. However, neuronal xCT activation also plays a significant role in GSH production, particularly under oxidative stress and neuroinflammation, where its expression is induced as a crucial self-defense mechanism. This is essential when astrocytic GSH supply is inadequate. In cancer research, the cystine transporter xCT forms a complex with the adhesion molecule CD44 variant (CD44v) to enhance GSH production and oxidative stress resistance (Figure 2a) [12]. Considering that CD44 is also a receptor for hyaluronic acid and CS [13,14], we hypothesized that the PG NCAN is involved in regulating the xCT/CD44 complex and the GSH regulatory mechanism.

We evaluated the effect of C4ST-1 expression on the CS modification of PGs by Western blot (Figure 2b). Using antibody anti-CS (1-B-5) antibody, which recognizes CS structures generated after chondroitinase (CSase) digestion (Figure 2b), we detected a major band in parental N2A cells at a position larger than 245 kDa, which corresponded to the core protein of NCAN. By contrast, in C4ST-1KO cells, this band intensity was significantly reduced. These results suggest that C4ST-1 loss reduces the CS modification of NCAN.

We then examined the complex formation of NCAN, xCT, and CD44v (Figure 2c). As NCAN is modified with CS chains, CSase digestion is required to visualize the NCAN core protein band. Immunoprecipitation with anti-xCT antibody from N2A cell lysate, followed by CSase treatment and Western blot, revealed a band of NCAN core protein at approximately 245 kDa. However, this band was not detected after siRNA-mediated NCAN knockdown. This finding indicates that NCAN binds to xCT possibly via CD44. Subsequently, immunoprecipitation of N2A cell lysates with anti-CD44 antibody showed that CD44v was detected at approximately 200 kDa, even without CSase digestion (Figure 2d). This result suggests that CD44v is not modified by CS chains in N2A cells, although CD44 is reportedly modified by CS [15]. By contrast, the protein level of CD44v was significantly reduced in C4ST-1KO cells compared with N2A cells (Figure 2d). Furthermore, xCT expression was significantly decreased in C4ST-1KO cells compared with N2A cells (Figure 2e). Likewise, in C17.2 cells, C4ST-1 deficiency significantly reduced xCT expression levels. These results suggest that C4ST-1 loss in neural cells reduces the CS modification of NCAN, which affects the stability of the NCAN/CD44v/xCT complex and may affect xCT-mediated GSH regulatory mechanisms through reduced xCT and CD44v protein levels.

### 3.4. C4ST-1 Deletion Alters Redox State and Promotes Protein Aggregation

Oxidative shifts in the intracellular environment cause protein denaturation and aggregation. Therefore, we investigated whether changes in the intracellular redox state affect protein aggregation in C4ST-1KO cells. We evaluated intracellular protein aggregation using ProteoStat fluorescent dye, an analog of thioflavin T (Figure 3a). Treatment with MG132, a proteasome inhibitor, to force aggregation increased the ProteoStat signal in N2A and C4ST-1KO cells. At steady state (untreated), the ProteoStat signal intensity in C4ST-1KO cells was significantly higher than that in N2A cells (Figure 3a). Similarly, in C17.2 cells, C4ST-1 deficiency increased the ProteoStat signal in the steady state (Figure 3a, bottom). This finding was similarly confirmed by flow cytometry (FACS) analysis (Figure 3b). These results suggest that C4ST-1 deficiency promotes intracellular protein aggregation even at steady state in the absence of stress.

We evaluated the effect of the accumulation of aggregating proteins on lysosomes (Figure 3c). Aggregated proteins are often difficult to be degraded by lysosomes, and their accumulation may damage lysosomal membranes [16]. Immunofluorescence staining with galectin 3, a marker of lysosomal membrane damage, showed significantly increased co-localization with the lysosomal marker LAMP1 in C4ST-1KO cells compared with N2A cells (Figure 3c). This result indicates that C4ST-1 deficiency promotes lysosomal membrane damage. To further examine the impairment of lysosomal function, we evaluated the sensitivity of lysosomes to the proton pump inhibitor bafilomycin A1. Compared with N2A cells, C4ST-1KO cells were significantly more sensitive to bafilomycin A1 (Figure 3d). These results suggest a series of pathological mechanisms by which the abnormal redox state caused by C4ST-1 deficiency promotes protein aggregation, leading to lysosomal membrane damage and dysfunction. This lysosomal dysfunction is consistent with a previous report that aggregating proteins such as α-synuclein and tau cause a vicious cycle in which lysosomal membranes are physically damaged, exacerbating tau aggregation via leakage into the cytosol [17].

### 3.5. C4ST-1 Deficiency Promotes Tau Protein Aggregation

Tau protein aggregation is a major pathological feature of AD and other tauopathies. Tau is a protein highly expressed primarily in the axons of neurons, where it binds to and stabilizes microtubules. This function is essential for maintaining cell morphology and axonal transport. In a pathological state, hyperphosphorylation of the tau protein occurs, causing tau to dissociate from the microtubules. Consequently, the microtubules become unstable, and axonal transport is impaired. Furthermore, the dissociated tau molecules abnormally aggregate with one another, forming insoluble fibrillar structures. This accumulation within the neuron constitutes neurofibrillary tangles. The accumulation of abnormal tau confers toxicity to the neuron, leading to cell death. Moreover, this pathogenic tau is thought to be released into the extracellular space, taken up by other healthy neurons as a seed, and subsequently propagate the pathology throughout the brain. The formation of this abnormal tau is crucial for the onset of tau pathology, and recent research reports that oxidative stress promotes tau aggregation through the formation of an intramolecular disulfide bond between two cysteine residues (Cys291 and Cys322) in the tau protein [18].

To evaluate the effect of oxidative stress caused by C4ST-1 deficiency on the disulfide bond formation of tau protein, we examined the oxidation state of thiol groups in tau using Protein-Shifter Plus (Figure 4a). In N2A and C4ST-1KO cells expressing wild-type tau protein, the thiol groups (-SH groups) of Cys291 and Cys322 were labeled with Protein-Shifter Plus. The results indicate that these cysteine residues do not form disulfide bonds and exist as reduced forms. We then analyzed the behavior of a tau mutant in which Cys291 and Cys322 were replaced with alanine. This mutant was not detected in N2A cells. However, a high-molecular-weight band was detected in C4ST-1KO cells. These results suggest that the mutant proteins degrade in N2A cells but form aggregates in C4ST-1KO cells. Finally, to evaluate the effect of C4ST-1 deficiency on tau pathology, we expressed tau protein with P301L, a mutation in familial tauopathy, and observed aggregation. An overview of the tau aggregation assay is shown in Figure 4b. Tau expressed in N2A and C4ST-1KO cells was fractionated and analyzed for both aggregated and soluble forms (Figure 4c). When tau aggregation was examined using AT8 antibody, which recognizes the hyperphosphorylated pathological tau protein, minimal aggregation occurred in both cells in the absence of tau seeds, a key agent responsible for tau aggregation. However, after the introduction of tau seeds, tau protein aggregation was significantly enhanced in C4ST-1KO cells compared with N2A cells. These results indicate that although reduced C4ST-1 expression alone does not cause tau protein aggregation, the intracellular environment created by C4ST-1 deficiency facilitates tau protein aggregation. In C17.2 cells, when a mutant tau was expressed with a seed, no phosphorylated tau aggregates were detected by AT8 (Figure 4d). However, when probed with a 4R-Tau antibody, non-phosphorylated tau aggregates of approximately 30 kDa were found in the C4ST-1KO C17.2 cells. This band was presumed to be a C-terminally truncated tau fragment because it was not recognized by the C-terminal-specific Tau46 antibody. As mentioned above, we have confirmed that C4ST-1 deficiency increases phosphorylated tau aggregates in N2A cells (Figure 4c). This difference is thought to be due to variations in the expression and activity of enzymes involved in tau phosphorylation and degradation between the two cell lines. Recent studies suggest that tau pathology is driven not only by hyperphosphorylation but also by protease-mediated cleavage [19]. Since tau fragments are more prone to aggregation than full-length tau, we propose that the oxidative environment caused by C4ST-1 deficiency promotes C-terminal tau cleavage, thereby enhancing tau’s aggregative properties. These results suggest that the oxidative stress resulting from C4ST-1 deficiency is a universal trigger for tau aggregation, even though the specific mechanism differs between cell types.

The fact that C4ST-1 deficiency alone did not induce tau aggregation does not weaken the significance of this study; rather, it strongly supports two key points. First, we can conclude that C4ST-1 is not a causative factor that initiates tau aggregation but an environmental factor that promotes the amplification of the pathology via oxidative stress. In cultured cells, the oxidative stress alone caused by C4ST-1 deficiency may not reach the threshold required to induce tau aggregation. However, since neurons in the body experience the long-term accumulation of aging and chronic inflammation, C4ST-1 deficiency can be considered to function as a factor that sensitizes or accelerates tau aggregation. Furthermore, AD and tauopathies are multi-stage pathologies that begin with neurons internalizing a tau seed; C4ST-1 deficiency was not directly involved in the formation of this tau seed. We propose that C4ST-1 deficiency plays a role in powerfully promoting the propagation process that occurs after a seed is introduced, by worsening the intracellular environment (oxidative stress, lysosomal dysfunction). These findings suggest that the reduction of C4ST-1 expression or CS 4-sulfation is not a factor that initiates tau aggregation but an environmental factor that determines the speed and toxicity of tau aggregation.

Interestingly, decreased expression of lysosomal arylsulfatase B (ARSB), responsible for the degradation of 4-sulfated CS, causes decreased intracellular GSH levels and increased oxidative stress [20]. This finding, which seems to contradict with the present study, suggests that CS 4-sulfation levels are tightly regulated within an optimal range for cellular homeostasis. A genome-wide association analysis has identified ARSB as a candidate gene for sporadic AD [21], and the expression of AD-related parameters is altered in the hippocampus and cortex of ARSB-deficient mice [22]. In addition, a longitudinal study in psychiatric patients identified ARSB as a novel blood biomarker for tracking short-term memory impairment, a key early symptom of AD [23]. The findings on C4ST-1 in the present study, together with the ARSB findings, suggest that maintaining proper CS sulfation levels is important to prevent oxidative stress and tau pathology in neurodegenerative diseases.

Although the association between human AD pathology and changes in C4ST-1 expression or CS 4-sulfation is unclear, studies using mice have shown the following findings. It has been reported that cognitive ability in the novel object recognition test is significantly reduced in very aged mice (30 months or older) compared to younger mice, and at the same time, C4ST-1 expression is also significantly reduced [24]. Furthermore, studies have shown that deleting C4ST-1 expression in adult mice leads to increased anxiety and abnormal social behavior, and these behavioral deficits are rescued by restoring C4ST-1 expression [25]. These preceding findings strongly suggest that changes in C4ST-1 expression and CS 4-sulfation may be linked to the development of AD in humans. Therefore, to connect the results found at the cellular level in this study to the elucidation of human disease, it is extremely important to investigate the expression of C4ST-1 and the state of CS 4-sulfation in both healthy individuals and AD patients, similar to the approaches taken in studies concerning ARSB.

## 4. Conclusions

Our study reveals a novel mechanism that C4ST-1 dysfunction disrupts cellular redox homeostasis and exacerbates tau pathology by reducing CS 4-sulfation. This disruption originates from the disruption of the neurocan/CD44v/xCT complex, which impairs GSH biosynthesis and induces oxidative stress. Thus, C4ST-1 and the specific CS structures that it produces may serve as new therapeutic targets for neurodegenerative diseases caused by oxidative stress.

## Figures and Tables

**Figure 1 cells-14-01686-f001:**
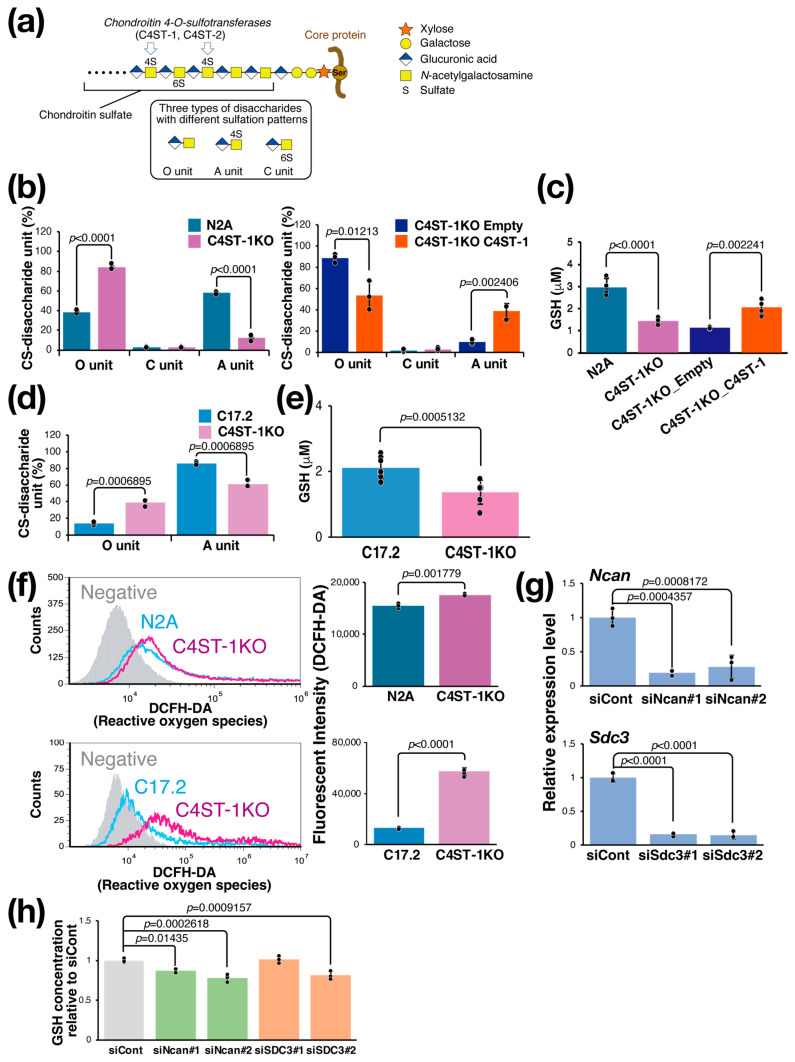
C4ST-1 loss reduces 4-sulfation, causing GSH depletion and oxidative stress. (**a**) Structure of PG and CS. The CS chain is composed of repeating disaccharides of glucuronic acid and GalNAc, which are sulfated in various patterns to create structural diversity. Three disaccharides with different sulfation patterns are shown in the black box. C4ST-1 is involved in sulfation of GalNAc at position 4. C4ST-1 acts on the O unit to synthesize the A unit. Symbols representing monosaccharides are shown on the right. (**b**) Disaccharides composition of CS isolated from N2A, C4ST-1KO, C4ST-1KO_Empty, and C4ST-1KO_C4ST-1 cells were analyzed. Data are presented as mean ± standard deviation (*n* = 3). Statistical analysis: Student’s *t*-test. (**c**) Intracellular GSH concentrations were measured in N2A, C4ST-1KO, C4ST-1KO_Empty, and C4ST-1KO_C4ST-1 cells. Data are presented as mean ± standard deviation (*n* = 4). Significance was determined by one-way ANOVA followed by Tukey’s multiple comparisons test. (**d**) Disaccharides composition of CS isolated from C17.2, and C4ST-1KO cells were analyzed. Data are presented as mean ± standard deviation (*n* = 3). Statistical significance was determined using the Student’s *t*-test. (**e**) Intracellular GSH concentrations were measured in C17.2 (*n* = 8), and C4ST-1KO (*n* = 7) cells. Data are presented as mean ± standard deviation. Statistical analysis: Student’s *t*-test vs. C17.2. (**f**) Intracellular ROS of N2A, N2A C4ST-1KO, C17.2, and C17.2 C4ST-1KO cells were measured by FACS using DCFH-DA as a probe. Negative indicates no DCFH-DA was added. The graph on the right quantifies fluorescence intensity. Data are presented as mean ± standard deviation (*n* = 4). Statistical significance was determined using the Student’s *t*-test. (**g**) Two siRNAs were introduced into N2A cells to knockdown Ncan (siNcan#1, siNcan#2) or Sdc3 (siSdc3#1, siSdc3#2), respectively. siCont is an siRNA sequence that does not target any gene product. Significance was determined by one-way ANOVA followed by Dunnett’s multiple comparisons test. (**h**) The intracellular GSH concentrations resulting from the knockdown of Ncan or Sdc3 were measured and expressed as a ratio of the GSH concentrations resulting from the siCont treatment. Data are presented as mean ± standard deviation (*n* = 4). Significance was determined by one-way ANOVA followed by Dunnett’s multiple comparisons test.

**Figure 2 cells-14-01686-f002:**
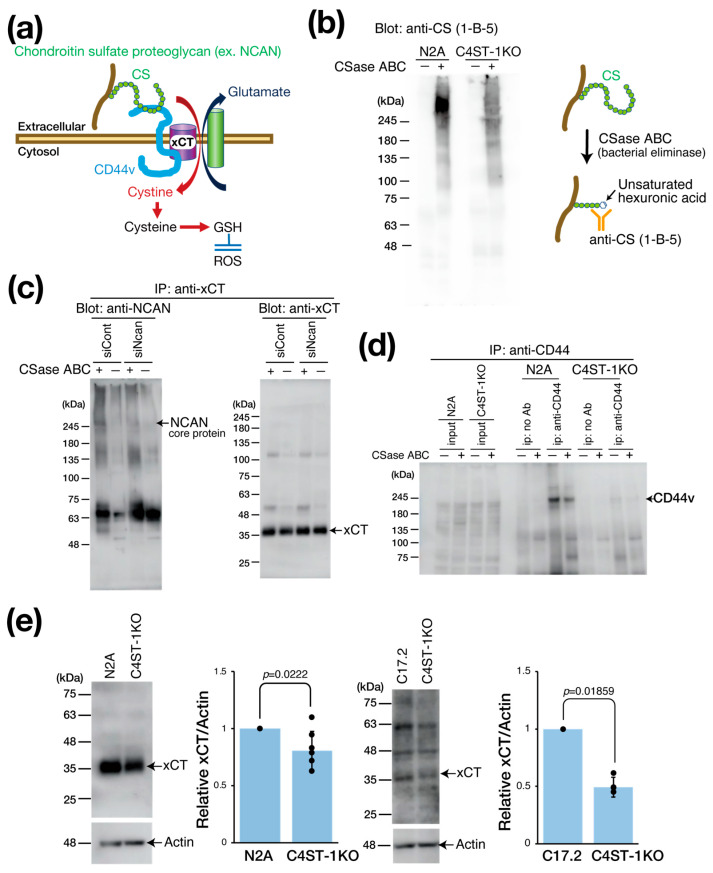
Effect of C4ST-1 deletion on xCT regulating intracellular GSH concentration. (**a**) Schematic diagram showing the xCT involved in the formation of GSH. CD44v interacts with and stabilizes xCT, a subunit of a glutamate-cystine transporter, and thereby promotes the uptake of cystine for GSH synthesis. CD44 binds CS as a ligand. (**b**) Cell lysates prepared from N2a and C4ST-1KO cells were digested with (+) or without (−) CSase ABC and subjected to Western blotting. The anti-CS (1-B-5) antibody used for detection recognizes the glycan structures that appear after CSase ABC digestion. (**c**) N2a cells were transfected with siCont and siNcan and immunoprecipitated with antibodies against xCT. Immunoprecipitates were treated with CSase ABC (+) or no treatment (−) and Western blotted with anti-NCAN or anti-xCT antibodies. (**d**) Cell lysates prepared from N2a and C4ST-1KO cells were immunoprecipitated with anti-CD44 antibody. Negative controls without antibody are indicated as no Ab. Immunoprecipitates were treated with CSase ABC followed by Western blotting with anti-CD44 antibody. (**e**) Cell lysates prepared from N2a (*n* = 7), N2A C4ST-1KO (*n* = 7), C17.2 (*n* = 5) and C17.2 C4ST-1KO(*n* = 5) cells were subjected to Western blotting using anti-xCT and anti-actin antibody. The graph on the right shows the expression level of xCT as a ratio to actin. Data are presented as mean ± standard deviation. Statistical significance was determined using the Student’s *t*-test.

**Figure 3 cells-14-01686-f003:**
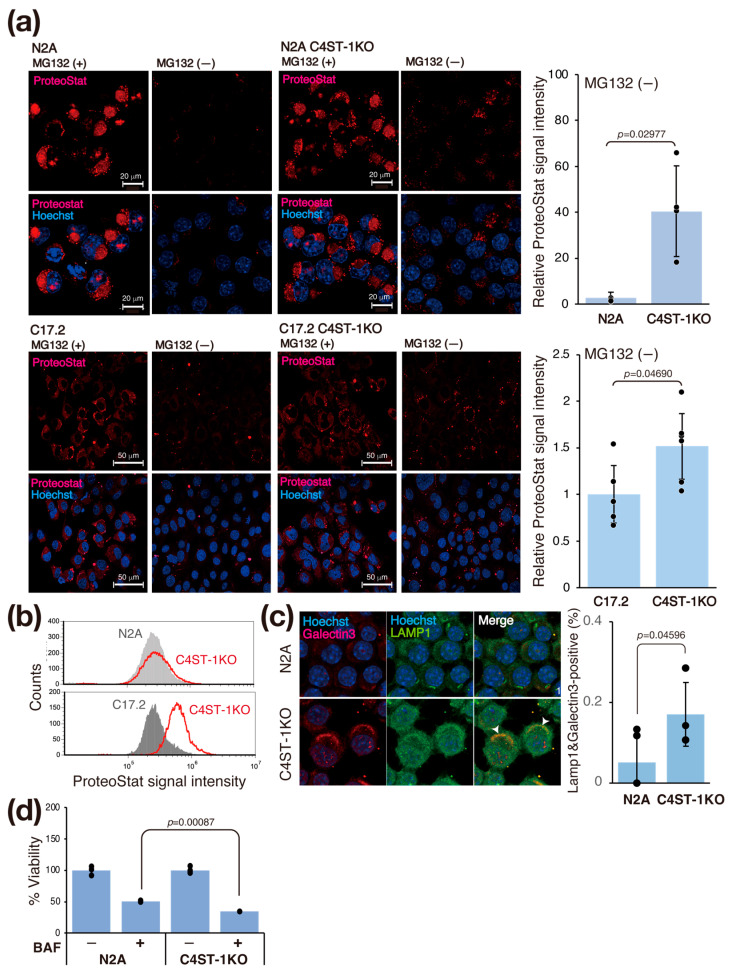
C4ST-1 deficiency promotes protein aggregation and impairs lysosomal function. (**a**) N2A, N2A C4ST-1KO, C17.2, and C17.2 C4ST-1KO cells fixed with 4% paraformaldehyde and permeabilized were stained with ProteoStat^®^ and Hoechst33342. The cells pre-treated with 5 mM MG132 for 6 h, a proteasome inhibitor, were used as a positive control. The graph on the right shows Proteostat fluorescence intensity divided by the number of nuclei. Data are presented as mean ± standard deviation (*n* = 4 or 5). Statistical significance was determined using the Student *t*-test. (**b**) N2A, N2A C4ST-1KO, C17.2, and C17.2 C4ST-1KO cells fixed with 4% paraformaldehyde and permeabilized were stained with ProteoStat^®^, and then analyzed by FACS. (**c**) N2A and C4ST-1KO cells were immunostained using anti-galectin-3 (red), anti-LAMP1 (green) antibodies, and Hoechst33342 (DNA, blue). The graph on the right shows the % of cells double-stained with galectin 3 and LAMP1 (arrowheads). Data are presented as mean ± standard deviation (*n* = 4). Statistical significance was determined using the Student *t*-test. (**d**) N2A and C4ST-1KO cells were treated with 10 nM bafilomycin A1 for 24 h, and then viable cells were measured by MTT assay. The viability of untreated C4ST-1 cells and bafilomycin A1-treated N2A and C4ST-1 cells are graphed, with the viability of untreated N2A cells set at 100%. Data are presented as mean ± standard deviation (*n* = 3). Significance was determined by one-way ANOVA followed by Tukey’s multiple comparisons test.

**Figure 4 cells-14-01686-f004:**
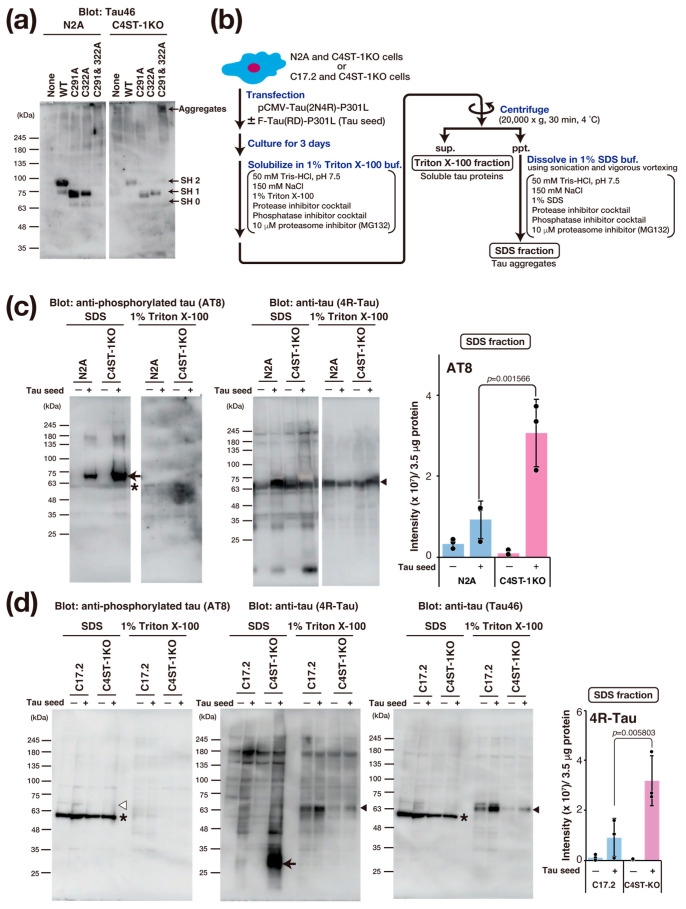
C4ST-1 deficiency accelerates the aggregation of a familial tauopathy mutant. (**a**) After expression of human wild-type Tau, C291A mutant, C322A mutant, and C291&322A mutant in N2A cells, the -SH group was labeled with Protein-Shifter Plus according to the manufacturer’s protocol. After electrophoresis, the large Protein-SHifter Plus moiety was eliminated with UV light (302 nm, 15 W, 10 min), increasing the efficiency of Western blotting and allowing for the detection of tau proteins using anti-Tau antibody (Tau46). Arrows indicate tau proteins with 0 to 2 free -SH groups and aggregates. (**b**) Cells were transfected with 2 µg of pCMV-Tau(2N4R)-P301L plasmid in the presence or absence of 4 µg of F-Tau(RD)-P301L. After culturing for three days, the cells were harvested in PBS, lysed with 1% Triton X-100 buffer, and then centrifuged. The resulting supernatant was designated as the Triton X-100 fraction, which contains soluble tau protein. The pellet was dissolved in 1% SDS sample buffer and designated as the SDS fraction, which contains tau aggregates. (**c**) N2A and C4ST-1KO cells were transfected with human tau mutant, Tau(2N4R)-P301L, together with (+) or without (−) tau seed, F-Tau(RD)-P301L, and cultured for 3 days. Cells were solubilized with 1% Triton X-100 buffer and then centrifuged. The supernatant obtained by centrifugation was used as 1% Triton X-100 fraction. The precipitates were solubilized in 1% SDS buffer to form the 1% Triton X-100 fraction and SDS fraction. Each fraction was analyzed by Western blotting using anti-phosphorylated tau antibody (AT8) and anti-tau antibody (4R-Tau). Arrow shows the phosphorylated tau aggregates, arrowheads indicate full-length tau, and asterisk denotes non-specific bands. The graph represents the signal intensity of phosphorylated tau protein bands in arbitrary units. Data are presented as mean ± standard deviation (*n* = 3). Significance was determined by one-way ANOVA followed by Tukey’s multiple comparisons test. (**d**) Tau aggregation was examined in C17.2 and C4ST-1KO cells using the same method described in (**c**). Tau proteins were detected using AT8, 4R-Tau, and Tau46 antibodies. Arrow indicates protease-cleaved tau fragments, arrowheads indicate full-length tau, and asterisks denote non-specific bands. White arrowhead corresponds to the migration position of phosphorylated tau aggregates, but no phosphorylated tau aggregates were detected in C17.2 cells.

## Data Availability

Data are available after communication with the corresponding author.

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
