# Peer review of "Dysfunctional Chondroitin 4-*O*-Sulfotransferase-1 Impairs Cellular Redox State and Promotes Tau Aggregation"

_cells, 2025, doi:10.3390/cells14211686_

Round 1

Reviewer 1 Report

Comments and Suggestions for Authors

This manuscript explores how dysfunction of chondroitin 4-O-sulfotransferase-1 (C4ST-1) disrupts neuronal redox homeostasis and enhances tau aggregation. The authors present convincing evidence that loss of C4ST-1 decreases chondroitin sulfate (CS) 4-sulfation, resulting in reduced glutathione (GSH) levels, elevated reactive oxygen species (ROS), and increased tau aggregation in neuronal cell models.

The research was carried out carefully using various appropriate methodologies.  Figures are clear and explanatory and the data is well presented and commented.

Minor comments:

Results relevance to human disease could be strengthened discussing potential alterations in C4ST-1 expression or CS 4-sulfation patterns in Alzheimer’s disease and other tauopathies, how it was done for ARSB, referencing available transcriptomic or proteomic datasets.

Correct  “Nagative” with “Negative” in Figure 1F.

Author Response

This manuscript explores how dysfunction of chondroitin 4-O-sulfotransferase-1 (C4ST-1) disrupts neuronal redox homeostasis and enhances tau aggregation. The authors present convincing evidence that loss of C4ST-1 decreases chondroitin sulfate (CS) 4-sulfation, resulting in reduced glutathione (GSH) levels, elevated reactive oxygen species (ROS), and increased tau aggregation in neuronal cell models.

The research was carried out carefully using various appropriate methodologies.  Figures are clear and explanatory and the data is well presented and commented.

Minor comments:

Results relevance to human disease could be strengthened discussing potential alterations in C4ST-1 expression or CS 4-sulfation patterns in Alzheimer’s disease and other tauopathies, how it was done for ARSB, referencing available transcriptomic or proteomic datasets.

Thank you for pointing this out. According to the reviewer’s comment, the following text has been added to lines 421-433.

Although the association between human AD pathology and changes in C4ST-1 expression or CS 4-sulfation is unclear, studies using mice have shown the following findings. It has been reported that cognitive ability in the novel object recognition test is significantly reduced in very aged mice (30 months or older) compared to younger mice, and at the same time, C4ST-1 expression is also significantly reduced (24). Furthermore, studies have shown that deleting C4ST-1 expression in adult mice leads to increased anxiety and abnormal social behavior, and these behavioral deficits are rescued by restoring C4ST-1 expression (25). These preceding findings strongly suggest that changes in C4ST-1 expression and CS 4-sulfation may be linked to the development of ADin humans. Therefore, to connect the results found at the cellular level in this study to the elucidation of human disease, it is extremely important to investigate the expression of C4ST-1 and the state of CS 4-sulfation in both healthy individuals and ADpatients, similar to the approaches taken in studies concerning ARSB.

Correct  “Nagative” with “Negative” in Figure 1F.

We have made the correction. Thank you.

Reviewer 2 Report

Comments and Suggestions for Authors
  1. The manufacturer and model information of the reagents used in the article also need to be listed in the attachment, such as cell culture.
  2. Discussion is needed to clarify the findings and significance of this paper, particularly regarding the C4ST-1 deficiency alone did not cause tau protein aggregation.
  3. For tau pathology, it did not present pathological results, it is recommended to provide.

Author Response

  1. The manufacturer and model information of the reagents used in the article also need to be listed in the attachment, such as cell culture.

Thank you for pointing this out. According to the reviewer’s comment, we added the manufacture and reagent information used in this study to supplementary table S3.

  1. Discussion is needed to clarify the findings and significance of this paper, particularly regarding the C4ST-1 deficiency alone did not cause tau protein aggregation.

Thank you for pointing this out. According to the reviewer’s comment, the following text has been added to lines 364-379.

The fact that C4ST-1 deficiency alone did not induce tau aggregation does not weaken the significance of this study; rather, it strongly supports two key points. First, we can conclude that C4ST-1 is not a causative factor that initiates tau aggregation but an environmental factor that promotes the amplification of the pathology via oxidative stress. In cultured cells, the oxidative stress alone caused by C4ST-1 deficiency may not reach the threshold required to induce tau aggregation. However, since neurons in the body experience the long-term accumulation of aging and chronic inflammation, C4ST-1 deficiency can be considered to function as a factor that sensitizes or accelerates tau aggregation. Furthermore, AD and tauopathies are multi-stage pathologies that begin with neurons internalizing a tau seed; C4ST-1 deficiency was not directly involved in the formation of this tau seed. We propose that C4ST-1 deficiency plays a role in promoting the propagation process that occurs after a seed is introduced, by worsening the intracellular environment (oxidative stress, lysosomal dysfunction). These findings suggest that the reduction of C4ST-1 expression or CS 4-sulfation is not a factor that initiates tau aggregation but an environmental factor that determines the speed and toxicity of tau aggregation.

  1. For tau pathology, it did not present pathological results, it is recommended to provide.

According to the reviewer’s comment, the following text has been added to lines 315-328.

Tau is a protein highly expressed primarily in the axons of neurons, where it binds to and stabilizes microtubules. This function is essential for maintaining cell morphology and axonal transport. In a pathological state, hyperphosphorylation of the tau protein occurs, causing tau to dissociate from the microtubules. Consequently, the microtubules become unstable, and axonal transport is impaired. Furthermore, the dissociated tau molecules abnormally aggregate with one another, forming insoluble fibrillar structures. This accumulation within the neuron constitutes neurofibrillary tangles. The accumulation of abnormal tau confers toxicity to the neuron, leading to cell death. Moreover, this pathogenic tau is thought to be released into the extracellular space, taken up by other healthy neurons as a seed, and subsequently propagate the pathology throughout the brain. The formation of this abnormal tau is crucial for the onset of tau pathology, and recent research reports that oxidative stress promotes tau aggregation through the formation of an intramolecular disulfide bond between two cysteine residues (Cys291 and Cys322) in the tau protein [20].